# Xenogeneic and Stem Cell-Based Therapy for Cardiovascular Diseases: Genetic Engineering of Porcine Cells and Their Applications in Heart Regeneration

**DOI:** 10.3390/ijms21249686

**Published:** 2020-12-18

**Authors:** Anne-Marie Galow, Tom Goldammer, Andreas Hoeflich

**Affiliations:** 1Institute of Genome Biology, Leibniz Institute for Farm Animal Biology, 18196 Dummerstorf, Germany; tom.goldammer@uni-rostock.de (T.G.); hoeflich@fbn-dummerstorf.de (A.H.); 2Molecular Biology and Fish Genetics Unit, Faculty of Agriculture and Environmental Sciences, University of Rostock, 18059 Rostock, Germany

**Keywords:** cell transplantation, myocardial infarction, mesenchymal stem cells, graft rejection, triple knockout pigs, genome editing, iPSCs, CRISPR/Cas

## Abstract

Cardiovascular diseases represent a major health concern worldwide with few therapy options for ischemic injuries due to the limited regeneration potential of affected cardiomyocytes. Innovative cell replacement approaches could facilitate efficient regenerative therapy. However, despite extensive attempts to expand primary human cells in vitro, present technological limitations and the lack of human donors have so far prevented their broad clinical use. Cell xenotransplantation might provide an ethically acceptable unlimited source for cell replacement therapies and bridge the gap between waiting recipients and available donors. Pigs are considered the most suitable candidates as a source for xenogeneic cells and tissues due to their anatomical and physiological similarities with humans. The potential of porcine cells in the field of stem cell-based therapy and regenerative medicine is under intensive investigation. This review outlines the current progress and highlights the most promising approaches in xenogeneic cell therapy with a focus on the cardiovascular system.

## 1. Cardiac Wound Healing and the Road to Xenogeneic Cell Therapy

There are two pathological events representing the clinically most relevant incidents in the cardiovascular system, namely rupture of an atherosclerotic plaque and myocardial infarction (MI). Both events are accompanied with severe tissue damage and loss of cardiomyocytes. The subsequent healing process is divided in two transitional phases. The early inflammatory phase is initiated by immigration of immune cells that secrete pro-inflammatory factors and clean out the tissue. An orchestra of neutrophils, monocytes, and lymphocytes acts for hours to days to remove necrotic tissue, phagocytize bacteria that may have settled, and release growth factors. The release of transforming growth factor beta (TGF-β), fibroblast growth factors (FGFs), and platelet-derived growth factor (PDGF) stimulates fibroblast proliferation, thereby inducing the following reparative phase. The reparative phase is characterized by enhanced matrix synthesis, proliferation of fibroblasts, and scar formation and usually lasts for days to weeks but may continue over years depending on the extent of the injury [1]. Dying cardiomyocytes secrete a variety of pro-inflammatory chemokines dedicated to evoke actions from bone marrow-derived cells and to attract immune cells. Understanding the role of stem cells in the modulation of these wound healing phases is of major relevance for the development of reparative therapeutics and stem cell-based therapies for cardiac repair.

The first early stage clinical trials on stem cell transplantation suggested beneficial effects on cardiac repair for both bone marrow [2,3] and cardiac-derived stem cells [4,5] although they were only modest. To avoid immunogenicity, these trials were mainly conducted with autologous cells. Due to the fact that autologous stem cells need to be expanded for up to three weeks before they can be applied in sufficient numbers for cell therapy [6], the respective cells were applied only after endogenous repair had begun. Hence, the beneficial effects might be attenuated after initialization of scar formation. It seems apparent that besides the current technological difficulties regarding proliferative capacity and phenotype maintenance, also the time requirements limit the clinical use of primary human cells. Together with the lack of human donors, this stimulated the search for alternative sources of cells. Pigs emerged as promising candidates for the production of donor tissues as they resemble many anatomical and physiological features of humans. For the cardiovascular system in particular, properties like an identical heart weight to body weight ratio, similar coronary circulation and hemodynamics, as well as comparable healing characteristics of the myocardium [7] rendered pigs not only as a suitable model organism but also as a potential donor for heart xenotransplantation.

Beginning in the 1990s, studies in diabetic animal models have demonstrated that porcine islet cell transplantation was sufficient to normalize blood glucose in the recipients, thus proving physiologic activity and metabolic regulation across the species barrier [8,9]. These findings raised hope for the implementation of xenogeneic cell replacement as potential therapy for a multitude of human diseases and disorders and inspired a number of research activities in the newly emerging field of xenogeneic cell therapy [10,11,12]. In fact, translation into clinical trials was achieved for the application of porcine islets of Langerhans with [13] and without Sertoli cells [14] to treat diabetic patients. Long-term follow-up studies documented decreased insulin requirements in a majority of patients [15] and the recent development of encapsulation strategies is supposed to overcome remaining immunological complications [16].

## 2. Overcoming the Immunological Barrier and Graft Rejection

In general, the transplantation of foreign tissue into a recipient with a functioning immune system will trigger an immunological reaction, which needs to be contained to prevent graft rejection. This is true for allografts, but it becomes even more considerable when the graft is of xenogeneic origin. There are several challenges introduced by the interspecies differences that will be discussed in the following.

### 2.1. Multilayered Immunological Challenges

The oligosaccharide galactose-α1,3-galactose (Gal) is present in all mammals except for humans and old world nonhuman primates [17]. Natural anti-pig antibodies to this carbohydrate antigen activate the complement-mediated immune response, resulting in the destruction of transplanted organs and tissues within minutes or hours by primarily targeting the vascular endothelial cells [18,19,20]. Causative factors of this hyperacute rejection were considered similar to ABO incompatibility in allograft rejection, thus similar measures were attempted to prevent rejection. Acceptance of ABO-incompatible renal allografts could be accomplished by extracorporeal immunoadsorption of anti-A or anti-B antibodies from recipient’s blood prior to the transplantation [21]. Accordingly, Taniguchi et al. depleted anti-Gal antibodies from baboon blood by specific immunoaffinity columns, thereby reducing serum cytotoxicity to pig kidney cells [22]. Thrombotic microangiopathy and subsequent graft failure can also be induced by activation of vascular endothelial cells by low levels of anti-nonGal antibodies or abnormal coagulation due to incompatibilities in the coagulation/anticoagulation factors [23,24].

For cellular xenotransplantation, the absence of vasculature attenuates the hyperacute xenograft rejection. However, there are additional immune responses that can be triggered by the xenogeneic cells, such as the activation of T-cells through direct and indirect pathways.

Direct activation is provoked by binding of recipient T cell receptors to swine leukocyte antigen class I and class II on porcine donor antigen-presenting cells, such as dendritic cells or endothelial cells constitutively expressing CD80/86 [25,26]. Indirect activation is initiated by the recognition of porcine donor peptides presented on the recipients major histocompatibility complex (MHC) class II. The subsequent T cell stimulation results in B cell activation and antibody production, thereby mediating humoral xenograft rejection [27].

Strategies to suppress xenoreactive T cells involved agents, such as CTLA4Ig (abatacept) and anti-CD40 mAb as well as anti-CD154 mAb, to block the T cell costimulation [28,29]. The soluble recombinant fusion protein, CTLA4Ig, attenuates costimulation mediated via CD28 by competing with its natural ligand B7 [30]. The costimulatory interaction of CD40 and its T cell-bound ligand CD154 is additionally involved in T cell-dependent B cell activity. Repeated administration of anti-CD154 mAb by intravenous injection could prevent an elicited antibody response to pig antigens in baboons [31]. However, combinatorial therapies targeting the CD28/B7 as well as the CD40/CD154 interaction represented the most effective approach for xenotransplantation as demonstrated for porcine skin [32], pancreas [33], and heterotopic heart transplants [34]. In all studies, agents were administered repeatedly every two to four days to maintain the effectiveness. Although these treatments improved xenogeneic graft survival to some extent, genetic engineering of the graft donor proved to be a far more successful approach.

### 2.2. Genetic Engineering to Overcome the Barriers

Pigs were genetically modified to render their tissues more resilient to the human humoral and cellular immune responses and to diminish the discrepancies to the human coagulation system. The first attempt was to introduce human complement-regulatory proteins, such as decay accelerating factor [35] and membrane cofactor protein (CD 46) [36], in porcine cells to protect them from the complement-mediated immune response of the recipient.

However, the breakthrough came only in the early 2000s when the first α1,3-galactosyltransferase gene-knockout (GTKO) pigs were generated by cloning [37] or nuclear transfer of spontaneous null mutant cells [38]. Knocking out the gene coding for the enzyme α1,3-galactosyltransferase prevents the production of Gal antigen, the main trigger for hyperacute rejections. Organs and cells from these GTKO pigs demonstrated prolonged graft survival. For example, heterotopically transplanted hearts were functional for more than two months in nonhuman primates, and in one animal, heart contractions ceased only after half a year. [39] However, xenograft rejection still occurred in GTKO pigs and was associated with activation of the innate immune system and coagulation [40]. Hence, to further improve the outcomes, GTKO pig cells were engineered to express human complement-regulatory proteins as investigated in vitro [41] and in vivo [42]. Whereas porcine endothelial cells transgenic for human complement regulatory protein CD46 were resistant to lysis by human and baboon serum [41], hearts from GTKO pigs expressing another complement regulatory protein CD55 did not show further extended graft survival in heterotopic cardiac pig-to-baboon transplantation, although local complement activation was restricted [42]. The observation of chronic vascular antibody deposition and endothelial cell activation suggested that rejection was initiated by non-Gal antigens.

In 2011, another porcine antigen, namely N-glycolylneuraminic acid (NeuGc), was found to elicit natural antibodies in humans, thereby increasing the transplant rejection risk [43]. NeuGc is expressed by most nonhuman mammals, which renders in vivo studies like in experimental pig-to-baboon organ transplantation problematic. However, the production of double knockout pigs by homozygous disruption of α1,3-galactosyltransferase (GGTA1), and cytidine monophosphate-N-acetylneuraminic acid hydroxylase (CMAH) prevents the adverse effects of these antibody–antigen interactions [44] and was a further step towards successful xenogeneic transplantation.

In 2015, the first triple knockout (TKO) pigs were created lacking the genes for GGTA1, CMAH, and additionally β4GalNT2, the gene coding for β1,4 *N*-acetylgalactosaminyltransferase (Sda), which was suggested to produce further xenoantigens for many humans [45]. Many humans display minimal or no natural antibodies to TKO pig cells, which leaves cellular rather than antibody-mediated rejection as the next hurdle for clinical application.

In order to address the T cell-mediated immune response, pigs overexpressing pCTLA4-Ig were produced both on a wild-type and GTKO background. However, these pigs exhibited reduced humoral immunity, which obliged the prophylactic use of antibiotics to promote their survival [46]. A promising alternative was proposed by the insertion of a mutant human MHC class II transactivator transgene in pigs that causes a downregulation of swine leukocyte class II expression and thereby reduces the human T cell response [47].

Another approach involves the overexpression of programmed cell death ligand 1 (PD-L1). In vitro studies with porcine PD-L1 expressing vascular endothelial cells and PD-L1 expressing B cells demonstrated a suppressed proliferation of human CD4 T cells and an enhanced expansion of regulatory T cells coupled with increased interleukin-10 production [48,49]. These results imply the potential of using human PD-L1-overexpressing porcine cells as a means to foster tolerance of other cell and tissue xenotransplants.

Human regulatory T cells can also be added directly in the context of xenotransplantation, where they have been shown to partly suppress the xenogeneic T cell response [50]. Besides regulatory T cells, mesenchymal stromal cells (MSCs) have unique anti-inflammatory and immunomodulatory properties and genetically modified porcine MSCs were shown to downregulate the human T cell response to porcine antigens in vitro, further implying considerable potential in xenotransplantation [51].

Complications with abnormal coagulation were tackled with additional genetic manipulations, such as the insertion of anticoagulant genes. Modification of GTKO pigs to express not only CD46 but also thrombomodulin resulted in prolonged graft survival [52]. Furthermore, porcine aortic endothelial cells transgenic for tissue factor pathway inhibitor were shown to exhibit a decreased expression of tissue factor, the primary initiator of blood coagulation [53].

In summary, the optimal xenogeneic donor might be a TKO pig that expresses minimally one human complement regulatory protein and another human coagulation regulatory protein. The introduction of further human transgenes, such as heme oxygenase 1 and CD47, might prove to be valuable for graft survival due to the general anti-inflammatory effects and the suppressive effect on monocyte and macrophage function, respectively [54]. Recent results with highly genetically engineered pigs were encouraging [55]; nevertheless, the study pointed at another pitfall, namely the influence of viruses [56].

## 3. Infection Risks in Xenogeneic Cell Therapy

The creation of TKO pigs constituted significant progress towards xenogeneic cell therapy. However, there are further risks and limitations associated with the transplantation of xenogeneic cells besides their immunological rejection. Complications during the engraftment, primary dysfunction of a graft, and potential transmission of infections are incidents to be considered attentively.

The infection risk associated with transplantation of porcine tissues into humans has been discussed already at the onset of research in this field [57,58] and is continuously curated today [59,60]. Hepatitis E virus (HEV), porcine cytomegalovirus (PCMV), porcine circovirus (PCV), and C-type porcine endogenous retroviruses (PERVs) are thought to pose a particular risk to possibly immunocompromised recipients. Detection systems for the respective viruses are discussed in a recent review [60].

HEV is a non-enveloped, single-stranded, positive-sense RNA virus that can induce hepatitis in humans. Besides two genotypes mainly found in humans, there are also two further genotypes (gt3 and gt4) that predominantly infect pigs [61]. However, HEV of the genotype 3 can be transmitted to humans by extensive contact with pigs or the consumption of undercooked pork [62]. Although the infections are asymptomatic in most cases and the rare symptomatic zoonotic diseases are usually mild, the virus might pose a risk in the setting of xenogeneic cell transplantations due to the potentially immunosuppressed status of the recipients. Therefore, different elimination strategies in herds generated for xenotransplantation are proposed.

The cytomegalovirus is an enveloped DNA virus belonging to the family *Herpesviridae*. For human cytomegalovirus, a negative impact on morbidity and mortality in allotransplantation is reported [63]. Thus, concern on the potential pathogenicity of porcine cytomegalovirus in xenotransplantation is deemed justified. The influence of PCMV on human transplant recipients is unclear. However, in baboons, PCMV transmission in orthotopic pig heart xenotransplantation was recently found to be associated with a shorter survival time of the transplant [64].

Circoviruses are non-enveloped spherical particles with a single-stranded circular small DNA genome belonging to the virus family *Circoviridae*. Porcine circovirus 3 (PCV3) is widespread among pigs and wild boars worldwide and was recently also detected in a herd of TKO pigs for xenotransplantation. After orthotopic heart transplantation, transmission of PCV3 to the recipient baboons was observed. A higher virus load in individuals with a longer survival time suggested an ongoing replication of the virus. However, in vitro experiments with 293T cells implicated that human cells could not be infected with PCV3 [65].

Of all the porcine viruses, PERVs are of particular concern due to the inherent characteristics of retroviruses. As endogenous viral elements in the genome, they cannot be eliminated by pathogen-free breeding. In contrast to human endogenous retroviruses that are in general inactive, PERVs have been shown to actively replicate in the pig [66]. Three closely related porcine endogenous retroviruses (PERV-A, -B, -C) were identified in pig [67], two of which are polytropic and can infect human cells [68,69]. Using clustered regularly interspaced short palindromic repeat-associated nuclease technology CRISPR/Cas, 62 PERVs could be inactivated in a porcine kidney cell line that normally releases high levels of infectious PERV in vitro [70]. Recently, the first PERV-inactivated pigs were successfully generated [71], suggesting that there might be completely PERV-free herds available for xenotransplantation in the future.

## 4. Stem Cells and Their Applications in Cardiovascular Regeneration

Stem cells can be subdivided in two categories, namely embryonic stem cells (ESCs) and adult stem cells. ESCs are derived from the inner cell mass of embryos in the blastocyst stage and can differentiate into any lineage. In contrast to adult stem cells, which demonstrate limited passage capacity, they can theoretically be maintained in culture indefinitely by continuous passaging. ESCs possess many further advantages, such as higher proliferation rate, stable morphology and telomerase activity under long-term cultivation, as well as pluripotency. Still, they turned out to be less favorable for cell-based therapies due to difficulties in efficient isolation, ethical concerns, and the widely reported tumorigenicity [72]. Therefore, research on clinical applications for cardiovascular diseases relies more heavily on adult stem cells nowadays.

A variety of adult stem cell types has been tested for their capacity for cardiac repair. A beneficial effect on cardiac function after injury was shown for cardiac progenitor cells [73,74], bone marrow-derived stem cells [75,76], induced pluripotent stem cells [77,78], as well as skeletal myoblasts [79,80]. These stem cell types display individual advantages, limitations, and practicability issues in clinical settings [81,82]. Progress in the application of different human stem populations as therapy for heart diseases in preclinical and clinical studies was recently evaluated [83].

Although stem cells derived from heart and bone marrow bear the capability to differentiate into the cardiac or mesenchymal lineage [84,85], it is now clear that they contribute minimally to newly formed cardiomyocytes [86,87,88]. Instead, stem cells induce neovascularization, stimulate endogenous stem cells, and modulate the post MI inflammatory response, thereby reducing scar formation [74,89,90].

### 4.1. Porcine Mesenchymal Stem Cells as a Promising Candidate in Cardiac Regeneration

Mesenchymal stem cells comprise a heterogeneous group of stromal cells holding postnatal capacity for self-renewal and multilineage differentiation [91]. They can be easily isolated from many tissues, such as bone marrow, adipose tissue, peripheral blood, and heart [92,93].

Due to their heterogeneity, a precise definition of MSCs remains problematic, but the International Society for Cellular Therapy defined minimal criteria to foster a more uniform characterization. Criteria are: (i) MSCs must be plastic adherent when maintained in standard culture conditions; (ii) MSCs must express CD105, CD73, and CD90, while lacking expression of CD45, CD34, CD14 or CD11b, CD79alpha or CD19, and human leukocyte antigen (HLA)-DR surface molecules; and (iii) MSCs must differentiate into osteoblasts, adipocytes, and chondroblasts in vitro [94].

MSCs are attractive candidates for cell-based therapy not only for their secretion of bioactive molecules activating tissue repair processes but also for their lack of cell surface histocompatibility complex (HLA) class II molecules and T cell costimulatory molecules, rendering them immunoprivileged. These characteristics together with a paracrine-mediated immunomodulatory activity allowed for the usage of allogeneic MSCs without pharmacological immunosuppression in cardiomyopathy patients [95], thereby raising hope for a successful application of porcine MSCs in cardiovascular regeneration across the xenogeneic transplant barriers. In fact, several studies provided evidence that MSCs successfully engrafted and functioned across the species barrier without any immunosuppression as lined out in the literature search of Li et al. [96]. Although the majority of studies concerned the transplantation of human MSCs in other species, there is also evidence for the successful engraftments of porcine cells into rodent brain [97], bone marrow [98], and infarcted hearts [99].

A comprehensive review on the characteristics of porcine mesenchymal stem cells and their therapeutic potential towards various diseases was provided by Bharli et al. [100]. For cardiovascular regeneration in particular, the administration of MSCs to diseased hearts results in improved cardiac function and reduced scar size. The underlying mechanisms for these positive effects are manifold, including direct and indirect contribution to angio- and arteriogenesis, regulation of immune responses, activation of cytoprotective pathways in reversibly injured cardiomyocytes, and stimulation of resident cardiac cell proliferation [101,102]. However, the relative contribution of each mechanism to the regenerative effect is yet to be illuminated. Although transdifferentiation of MSCs into cardiomyocytes and vascular components was repeatedly described [103,104]; evidence consolidates that the contribution of MSCs to heart regeneration is mainly based on their release of numerous cytokines and paracrine factors, rather than direct cell replacement [89,105,106]. Allogeneic cell combination therapy utilizing not only mesenchymal stem cells but also cardiac stem cells was proven to elicit synergistic effects [107] and might also prove beneficial in prospective xenogeneic therapies. In fact, studies with human bone marrow-derived MSCs and c-kit-positive cardiac stem cells demonstrated that applications across the human–pig species barrier are feasible [108,109]. Table 1 presents a list of studies utilizing porcine mesenchymal stem cells in diverse disease models to explore their effect on cardiac regeneration. Notably, porcine mesenchymal stem cells were primarily applied in autologous or allogenic approaches, reflecting the significance of the pig as a preferential preclinical model for human cardiac pathologies but leaving open the question why there are no pig-to-baboon studies for stem cell transplantation. In fact, preclinical studies on stem cell therapy for cardiac diseases in non-human primates are scarce in general [110] and concentrated on human embryonic stem cell-derived cardiomyocytes [111,112] or iPSCs [113]. Thus, the lack of xenogeneic studies on porcine mesenchymal stem cells is most likely attributable to a combination of preference of the pig over the mouse as a preclinical model system and rareness of ethically more critically considered primate studies.

### 4.2. Delivery and Final Fate

Cell dosing and the route of administration are crucial for the outcome of cell-based therapies. However, missing standards are still leading to inconsistent results in preclinical and clinical studies on stem cell-based therapy for cardiovascular disease [125]. Implementing efficient delivery methods and assuring long-term retention of grafted cells in the target area is challenging. There are a variety of methods for cell administration into a patient’s myocardium, such as direct surgical intramyocardial injection, catheter-based intramyocardial administration, intravenous infusion, intracoronary artery administration, retrograde coronary venous delivery, and transplantation of engineered monolayer tissue [126], each with individual advantages and drawbacks.

Despite the availability of safe and technologically feasible delivery routes, benefits in cardiac function restoration observed in preclinical animal models could not be reproduced in humans so far. One underlying reason is that all delivery routes demonstrate rather low ratios of cell engraftment and survival in preclinical and clinical trials.

For example, early studies in an ischemic swine model demonstrated retention rates of peripheral blood mononuclear cells within the myocardium of only about 2.6%, 3.2%, and 11% after intracoronary, interstitial retrograde coronary venous, and intramyocardial delivery, respectively [127]. For these low retention rates, some of the delivery routes are controversially discussed regarding their actual impact on the recovery of cardiac function [126,128].

As a consequence, an increasing number of studies nowadays focuses on combining stem cells with synthetic or natural scaffolds. For cell delivery, both solid scaffolds used as biomimetic cardiac patches as well as soluble materials, such as injectable hydrogels, are commonly applied [129,130]. In general, these matrices are supposed to provide a suitable cellular microenvironment and ideally allow mechanical and electrical coupling with the host myocardium.

For preclinical trials, swine is commonly used as the translational model, for example, for the application of decellularized extracellular matrices (ECMs) of porcine as well as human origin as a scaffold for stem cells. Such bioactive constructs have been demonstrated to integrate well into cardiac tissues, where they support cardiac function and reduce myocardial scar formation [117,131,132]. However, decellularization of porcine heart tissue followed by efficient recellularization with patient-specific human induced pluripotent stem cells (iPSCs) to gain a fully functional cardiac patch poses a breakthrough that has yet to be achieved.

Interestingly, significant therapeutic effects in postischemic models were also observed when such constructs were used without cells. It is now understood that ECM can support cardiac regeneration by stimulating angiogenesis and cardiomyocyte proliferation via biochemical and biomechanical signaling [133,134,135]. Naturally occurring sources of suitable extracellular matrices include small intestinal submucosa, urinary bladder matrix, pericardium, as well as myocardial tissue itself [136].

### 4.3. Genetic Engineering to Improve Efficiency of Mesenchymal Stem Cells

Several mechanisms may account for the poor engraftment and survival rates of transplanted stem cells. During preparation of the cell suspensions, proapoptotic signaling might be triggered by compromised integrin-mediated survival and discontinuation of contacts and intercellular communication between cells and the extracellular matrix. After transplantation, apoptosis could be activated by several stressors present in the infarct and/or peri-infarct area, such as hypoxia, acidosis, nutrient deficiency, reactive oxygen species (ROS), inflammatory mediators, and cytotoxic agents [101]. The survival of transplanted MSCs is determined by the balance between these proapoptotic signals, mainly mediated through toll-like receptor 4 (TLR4) and G protein-coupled receptors, on the one hand and prosurvival pathways, such as the phosphoinositide 3-kinase (PI3K), protein kinase B (AKT), and extracellular signal-regulated kinase 1/2 (ERK1/2) pathways, on the other hand.

To increase the therapeutic effect of MSCs, several attempts have been made to render them more resistant to apoptosis. Targeted deletion of the proapoptotic TLR4 resulted in increased AKT activation and decreased hypoxia-induced apoptosis in murine MSCs [137] associated with improved myocardial recovery by elevated angiogenic factor production [138]. On the other side, porcine bone marrow-derived MSCs were transduced with AKT to strengthen prosurvival signals [139]. Indeed, AKT-MSCs showed reduced intracellular ROS levels and apoptosis induced by H_2_O_2_. Moreover, they demonstrated higher levels of vascular endothelial growth factor (VEGF) production and ERK1/2 activation, resulting in improved cardiac function after cellular transplantation.

Other genetic modifications for increased MSC survival target the heme oxygenase-1 gene; heat shock proteins; antiapoptotic proteins, such as Bcl-2, Bcl-xL, connexin 43, and survivin; or transcription factors, such as GATA-4 and HIF-1α, but are mainly conducted in rodent models and not yet translated to preclinical large animal models [101].

Retention could be improved by manipulation of the integrin signaling pathway. Porcine MSCs overexpressing integrin-linked kinase showed increased proliferation and reduced apoptosis, demonstrated by the TUNEL method (terminal deoxynucleotidyl transferase-mediated dUTP nick-end labeling). Upon transplantation, enhanced angiogenesis and cardiomyocyte proliferation was observed in pigs after MI while fibroses was reduced, resulting in improved ventricular remodeling and cardiac function [140].

Next to improving the retention and survival rates, there is another approach to improve the efficiency of stem cells, namely to increase their production of cardioprotective cytokines and growth factors. The most relevant factors for myocardial regeneration secreted by MSCs include VEGF, TGFβ, hepatocyte growth factor (HGF), platelet-derived growth factor (PDGF), stromal cell-derived factor 1 (SDF-1), fibroblast growth factor β (FGFβ), angiopoietin 1 (ANG-1), granulocyte colony-stimulating factor (G-CSF), and interleukin-8 (IL-8) [141].

The overexpression of both insulin-like growth factor 1 (IGF-1) and HGF in MSCs resulted in reduced inflammation and improved angiogenesis in a pig MI model. However, cardiac function parameters were not significantly improved [116]. In contrast, intramyocardial delivery of bone marrow mesenchymal stromal cells overexpressing mutant human HIF1-α reduced infarct size and improved LV systolic performance in another large animal model [142]. VEGF and ANGP-1 were significantly upregulated in the peri-infarct zone, supporting increased neovascularization and cardioprotective effects. Hypoxia-induced overexpression of VEGF facilitated its ischemia-responsive production and was sufficient to significantly increase myocardial neovascularization and attenuate left ventricular remodeling after myocardial infarction in a rat model [143].

The recent development and refinement of programmable nucleases, such as zinc-finger nucleases, transcription activator–like effector nucleases (TALENs), and clustered regularly interspaced short palindromic repeat (CRISPR)–Cas nucleases, may soon allow for a save application of gene editing in clinical practice. High nuclease activity and specificity was proven in various preclinical disease models and a number of clinical gene-editing trials are on the way [144].

However, despite the progress in DNA-based approaches, miRNA approaches might pose the more applicable method for future clinical translation as epigenetic modifications hold the possibility of transient action. There are three main strategies pursued with miRNAs: (i) improving the survival rates of transplanted cells, (ii) enhancing therapeutic efficacy by increased factor secretion, and (iii) supporting differentiation in cardiomyocytes.

The ability of miRNAs to protect MSCs during post-transplantation in the infarct heart has been demonstrated for miR-133 [145], miRNA-301a [146], and a cocktail of miR-21, miR-24, and miR-221 [147]. By employing miR-126 [148] or miR-146 [149], stem cells could be modified to enhance their paracrine release of angiogenic factors, such as VEGF, thereby improving ischemic angiogenesis and cell survival in the damaged heart. For the promotion of cardiomyocyte differentiation, overexpression of let-7 family members was shown to result in a matured morphology as well as increased force of contraction and respiratory capacity in human stem cell-derived cardiomyocytes [150]. Furthermore, overexpression of miR1-2 has been shown to promote cardiomyocyte differentiation in mouse MSCs via the activation of the Wnt/β-catenin signaling pathway [151]. Targeting the same pathway, also overexpression of miR-499 could induce mesenchymal stem cells toward cardiac differentiation [152]. Recently, first therapeutic miRNA-based applications inhibiting miR-132 (NCT04045405) or miR-92a (NCT03603431) were examined, marking the onset of translation of cardiac epigenetic modifications into the clinic.

The genetic engineering approaches in mesenchymal stem cell therapy as well as the other earlier mentioned xenogeneic approaches in heart regeneration therapy are illustrated in Figure 1.

## 5. Future Prospects for Xenogeneic Cell Therapy

Allogenic cell transplantation as therapy to improve cardiac function was demonstrated to be feasible in pig [153] but also in human [154], where the administration of allogenic cells did not trigger immune-mediated events. Thus, one could question why xenogeneic cell therapy should be considered at all when allogeneic cell transplantation is possible, but there are major advantages coming with the pig as donor.

Even if one were to leave aside the lack of human donors, there is one basic and initiatory obstacle to the broad deployment of stem cell-based therapy. The low isolation rates of these primary cells cause the need for expansion before an application in the patient is appropriate. In clinical studies applying MSCs for the treatment of chronic and acute cardiovascular diseases, high cell numbers, ranging from 1 × 10^6^ up to more than 1 × 10^9^, are typically administered to compensate for the low survival and retention rates [155]. The time lapse between stem cell isolation and delivery renders early engraftment in acute MI impossible when relying on autologous cells or freshly isolated cells of allogenic donors.

Cryopreservation could bypass this bottleneck and allow for readily available cells, as it is the gold standard when cell storage and transportation is required. However, there are deviating observations regarding the impact of cryopreservation on various MSC features as discussed in detail in a recent review [156]. As opposed to physical damage, other aspects, such as gene expression alterations, stress responses, and epigenetic, changes after thawing are usually less assessed. The present data suggest that cryopreservation has no significant influence on proliferation, morphology, or differentiation of bone marrow-derived MSCs but does appear to influence viability, attachment, immunomodulation, and metabolism.

Since there is now general consent that the positive effect of transplanted MSCs in cardiac regeneration is mainly attributed to paracrine signaling instead of differentiation into cardiomyocytes, stable proliferation and differentiation potential after cryopreservation may not represent the adequate features for evaluation of their functional potency.

In contrast to human donors, genetically modified pig lines with minimal immunogenicity are constantly present and porcine stem cells of various origin can be harvested in advance at all times.

Porcine stem cells have proven their potential for cell-based therapy in cardiovascular diseases in many studies as discussed in this review. Furthermore, the possibility to genetically modify pigs is a major advantage, which would be difficult to transfer into the clinic with allogeneic human cells. A variety of genome engineering strategies have been employed in context with xenotransplantation [157], not only to delete porcine genes or insert human genes, but also to engineer stem cells with optimized therapeutic effects. In contrast to ESC-based genome editing in mice, these strategies usually rely on somatic cell nuclear transfer of primary cells, which is more complex and elaborate [158]. The lack of porcine ESC stocks results in lower efficiencies and higher costs for the generation of modified pigs. The establishment of porcine ESCs for genetic manipulation would advance the future prospects of xenogeneic cell therapy and should be an integral part of upcoming work in the field.

The advantages of broad availability and genetic modifiability are recently also shared with human-induced pluripotent stem cells (iPSCs). Allogeneic iPSCs banking is envisioned to provide off-the-shelf treatment options and first clinical trials already included their application in heart failure patients (NCT03763136), but these cells are still associated with some drawbacks.

While porcine MSC lack HLA class II molecules and are naturally immunoprivileged, the individual MHC haplotypes need to be considered for allogeneic iPSCs to avoid the T cell-mediated immune response and rejection of grafted cells. Interestingly, in murine models, even autologous iPSCs were shown to elicit an immune response when used in an undifferentiated state [159].

Generating one clinical-grade iPSC line is associated with financial expenses around 800,000USD [160]. Thus, instead of generating fully personalized iPSCs, HLA-haplotyped iPSC cell banks matching to the majority of recipients are discussed as an alternative approach [161]. The establishment of such cell banks is not free of barriers and the costs are still considerable [160,162], rendering the application of iPSCs for the production of cellular therapeutics commercially not necessarily superior to porcine MSCs. Furthermore, there is still some uncertainty on the actual impact of reported genetic aberrations in iPSCs in terms of functionality and the risk of teratoma formation [163].

On the other side, the size of adult pigs represents a husbandry challenge for standard laboratory facilities and costs for housing are substantial. For this reason, the generation of porcine MSCs might remain restricted to specialized laboratories, limiting the options for acute applications without employing potentially adverse cryopreservation for transportation. Both allogeneic and xenogeneic cell therapy approaches have their limitations. The future will show which strategy will be more feasible for applications in clinical settings.

Recently, genome editing by RNA-guided endonucleases (CRISPR/Cas9) in combination with iPSCs has paved the way for many exciting new opportunities, one of which is the creation of human cells and even organs from genetically modified chimeric pigs. In 2017, a pig blastocyst was successfully injected with human cells to create a chimeric pig for the first time [164]. Although chimeric pigs might provide an unrestricted resource of cells and organs that would be completely tolerated by the recipient’s immune system, there are also fundamental ethical concerns [165].

An ethically uncontroversial approach is the application of decellularized tissues and matrices for cardiac regeneration therapy [166] as mentioned earlier. For this, decellularized extracellular matrix derived from porcine myocardium turned out to be the ideal scaffold for cardiac patches [167] or as a source for injectable matrix hydrogels [168], both shown to improve cardiac repair after myocardial infarction. Due to the limited access to healthy human hearts as a source for ECM, age-dependent shifts in ECM composition and significant patient-to-patient variability, human decellularized myocardium is not a well scalable option for the clinic. Thus, the pig remains as an optimal donor and xenogeneic therapy as a prospective clinical approach for heart regeneration. In 2019, a clinical trial on the application of a hydrogel derived from decellularized porcine myocardium proved its safety and feasibility and suggested improvements in 15 early and late myocardial infarction patients, warranting validation of efficacy in larger trials [169].

As research on cell therapies focuses more and more on iPSCs, the future of xenogeneic therapy approaches could shift towards acellular products. Either way, there will be special regulations to meet. Cell-based medicinal products, which apply also to xenogeneic products, are subject to specific regulations and directives guided by agencies, such as the US Food and Drug Administration or the European Medicines Agency [170]. Continuous monitoring for infectious pathogens in designated pathogen-free source animals must be accompanied by a standardized storage of samples and archiving of records from the donor animal. Acellular products like heart valves and other decellularized matrices are not classified as cell therapy products but are subject to their own regulations. Documents about safety and requirements for entering clinical trials are provided by the WHO and the International Xenotransplantation Association [144]. With appropriate regulatory frameworks in place, the way should be paved for xenogeneic cell therapy for cardiovascular diseases, in particular utilizing MSC and ECM products.

## Figures and Tables

**Figure 1 ijms-21-09686-f001:**
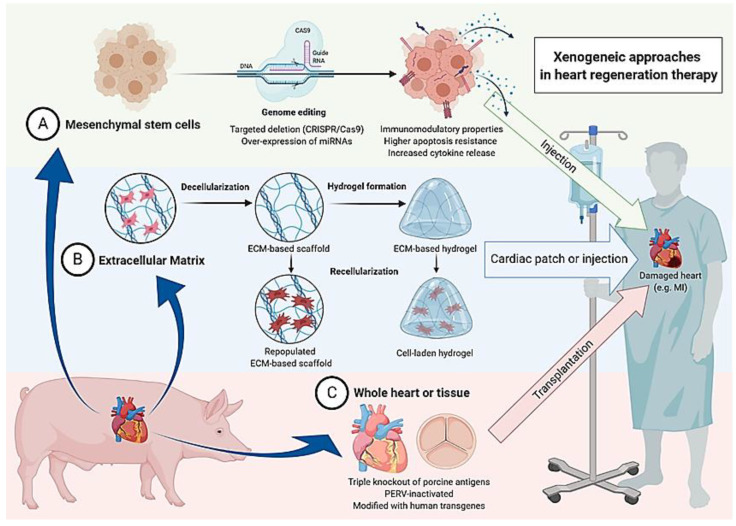
Summary of three conceivable xenogeneic approaches in heart regeneration therapy. Generally possible applications utilizing the pig as a donor include (**A**) transplantation of mesenchymal stem cells, (**B**) insertion of acellular or populated extracellular matrix, and (**C**) transplantation of the whole heart or heart valves of genetically engineered animals.

**Table 1 ijms-21-09686-t001:** Studies on the effect of porcine mesenchymal stem cells in heart regeneration. The table outlines the conditions and outcomes of mesenchymal stem cell-based therapies in several cardiovascular disease models published 2010–2020. ATMSCs: Adipose tissue-derived mesenchymal stem cells, ATSCs: adipose tissue-derived stem cell, CDCs: Cardiosphere-derived cells.

Cell Type and Cell Number	Delivery Method	Clinical Condition	Main Outcome	Study
Autologous MSC 3 × 10^7^ cells	Direct left ventricular injection	Acute MI	Improved cardiac function Reduced oxidative stress and inflammatory response	[114]
Allogeneic ATMSCs 1 × 10^7^ cells	Intracoronary infusion	Acute MI	Unaltered cardiac function Unaltered scar size Promoted revascularization	[115]
Allogeneic ATMSCs 5 × 10^7^ cells Overexpressing IGF-1 or HGF	Intramyocardial injection	Acute MI	Unaltered cardiac function Promoted revascularization Reduced inflammation Enhanced fibrosis	[116]
Allogeneic ATDPCs 1.75 × 10^6^ cells	Recellularized myocardial scaffold	Acute MI	Improved cardiac function Reduced infarct size Promoted revascularization Attenuated fibrosis	[117]
Xenogeneic MSCs 1 × 10^6^ cells	Intramyocardial injection	Acute MI	Improved cardiac function Unaltered infarct size Improved preservation of capillarity	[99]
Allogeneic MSCs 1.4 × 10^7^ cells	Intramyocardial injection	Subacute MI	Unaltered cardiac function Reduced scar size	[118]
Autologous BMCs eNOS-transfected 1 × 10^7^ cells	Intramyocardial injection	Subacute MI	Improved cardiac function Unaltered scar size Unaltered fibrosis	[119]
ATSCs 1 × 10^7^ cells	Intramyocardial injection	Subacute MI	Improved cardiac function Reduced infarct size	[120]
Autologous MSC 6 × 10^7^ cells	Intramyocardial injection	Chronic MI	Unaltered cardiac function Unaltered scar size Attenuation of LV wall thinning	[121]
Autologous BMCs 5 × 10^6^	Left or right coronary artery infusion	Chronic MI	Improved cardiac function Promoted revascularization	[122]
Allogeneic MSC 2 × 10^8^ cells	Intramyocardial injection	Chronic MI	Improved cardiac function Reduced scar size	[104]
Allogeneic MSC 1.5 × 10^7^ cells	Recellularized collagen scaffold	Chronic MI	Improved cardiac function Reduced scar size Promoted revascularization	[123]
Allogeneic MSCs and CDCs 3.5 × 10^7^ cells	Intracoronary infusion	Chronic ischemic cardiomyopathy (coronary stenosis)	Improved cardiac function Unaltered perfusion Improved remote wall thickening	[124]
Allogeneic MSCs and CSCs 2 × 10^8^ cells	Transendocardial injection	Chronic ischemic cardiomyopathy	Improved cardiac function Reduced scar size Low-grade inflammatory infiltrates	[107]

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
