# Peer review of "Xenogeneic and Stem Cell-Based Therapy for Cardiovascular Diseases: Genetic Engineering of Porcine Cells and Their Applications in Heart Regeneration"

_ijms, 2020, doi:10.3390/ijms21249686_

Round 1

Reviewer 1 Report

This paper addresses the potential use of engineered porcine stem cells for cardiovascular "regeneration". This topic has been around for many years and is clearly of interest. Unfortunately, this article is fraught with several issues.

1- The major concern is that only a limited fraction of the paper is really devoted to the announced topic. From section 4, the paper is rather a general review on stem cells, primarily mesenchymal stem cells (MSC). This is the case for section 4.1 in which a table summarizes studies on the effects of porcine MSC but only comprises one reference about xenogeneic cells (the purportedly main subject of the paper) while the others deal with auto- or allogeneic cell sources. Likewise, the following sections on delivery of cells and their genetic engineering are general considerations, irrelevant to the subject and mostly reiterating comments that have already been repeatedly published.

2- At the beginning of the paper, the authors only briefly allude to the use of encapsulated xenogeneic pig islets for treating diabetes. This section should have been expanded to review the most recent developments, both experimental and clinical, as this strategy remains so far the most advanced translational example of xenogeneic cell transplantation.

3- To support the assumption that MSC can function across the species barrier, the authors quote a paper (ref 83) which is a general review in which 89 out of 94 reports involved the use of human MSC in various other species, with only 2 pig-to-mouse (of which one entailed co-transplantation of MSC with hematopoietic stem cells) and one pig-to-rat experiments. It is therefore questionable to draw meaningful conclusion regarding the relevance of porcine stem cells to the overall field of cell transplantation from this study and a much more comprehensive review of the potential of porcine MSC has actually been provided by the paper by Bharli et al. (Curr Stem Cell Res Ther. 2016 Jan; 11(1): 78–93.) .

4- More importantly, the case is made that porcine MSC could have the advantage of an unlimited scale-up. The authors state that "porcine stem cells of various origin can be harvested in advance at all times". While this is true, this advantage is now shared by pluripotent stem cells and, more particularly, induced pluripotent stem cells whose growth potential and suitability for large-scale banking challenge those of porcine cells (while largely eliminating the risks of pig-specific infection). Unfortunately, this issue is not addressed in the manuscript.

5- The authors also state that in the case of acute myocardial infarction, time constraints result in that one cannot rely on, "freshly isolated cells of allogeneic donors". This is not true as cryopreserved allogeneic cells can be rapidly thawed and used.

6- It is also stated that the porcine extracellular matrix could be "ideal as scaffold for cardiac patches". However, there is no mention of the clinical trial which has tested this strategy (Traverse et al. JACC Basic Transl Sci. 2019 Oct; 4(6): 659–669.).

Author Response

Point 1: The major concern is that only a limited fraction of the paper is really devoted to the announced topic. From section 4, the paper is rather a general review on stem cells, primarily mesenchymal stem cells (MSC). This is the case for section 4.1 in which a table summarizes studies on the effects of porcine MSC but only comprises one reference about xenogeneic cells (the purportedly main subject of the paper) while the others deal with auto- or allogeneic cell sources. Likewise, the following sections on delivery of cells and their genetic engineering are general considerations, irrelevant to the subject and mostly reiterating comments that have already been repeatedly published.

Response 1: We acknowledge the reviewers critical comments. Since the title of the review may have raised wrong expectations, we have corrected it to “Xenogeneic and stem cell-based therapy for cardiovascular diseases: genetical engineering of porcine cells and their applications in heart regeneration”.  We feel that the new title better indicates the strong emphasizes on (mesenchymal) stem cells and thus contributes to a higher degree of clarity.

In addition, we would like to give a clarifying statement on the aim of this article. Our review deals with the emerging role of porcine cells in cardiovascular regeneration. This emerging role is embedded in a broader context. As interpreted by Reviewer 3; “This review can be informative to the readers who seek an overview of porcine xenotransplantation products including MSCs and their therapeutic applications.” A similar and recent review is not available at present. We believe that the topic is highly relevant, positioned in a dynamic field of applied science and importantly, is addressing also readers less experienced in the field. We therefore consider the review as an important contribution.  In this context and opposed to the reviewer’s statement, we believe that the sections on delivery of cells and engineering are indeed relevant to give a comprehensive overview on all aspects of proposed cell therapy approaches.

We ourselves think the lack of xenogeneic studies in the table is unfortunate. However, as now stated and discussed in lines 274-282 there simply are no more studies on porcine MSCs in xenogeneic settings we could have referred to.

Point 2: At the beginning of the paper, the authors only briefly allude to the use of encapsulated xenogeneic pig islets for treating diabetes. This section should have been expanded to review the most recent developments, both experimental and clinical, as this strategy remains so far the most advanced translational example of xenogeneic cell transplantation.

Response 2: As suggested by the reviewer, we added a comment on the success of xenogeneic islets for diabetes treatment in lines 62-66 including several respective references for the interested reader. However, since the scope of this review is on applications in cardiovascular research and heart regeneration we would prefer not to expand this section further.

In fact, the use of xenogeneic pig islets for treating diabetes has been reviewed very frequently in the last years, four times just in 2014 (see Samy et al. Islet cell xenotransplantation: a serious look toward the clinic. Xenotransplantation. 2014. doi: 10.1111/xen.12095.;
Zhu et al. Pig-islet xenotransplantation: recent progress and current perspectives. Front Surg. 2014. doi: 10.3389/fsurg.2014.00007.;

Reichart et al. Xenotransplantation of porcine islet cells as a potential option for the treatment of type 1 diabetes in the future. Horm Metab Res. 2015. doi: 10.1055/s-0034-1395518.;

Hu et al. Pig islets for islet xenotransplantation: current status and future perspectives. Chin Med J (Engl). 2014;127(2):370-7. ) and the last time in April this year (see Matsumoto et al. Current situation of clinical islet transplantation from allogeneic toward xenogeneic. Journal of Diabetes. 2020; doi.:10.1111/1753-0407.13041.

With this review we aim to provide a comprehensive overview in the light of a less frequently reviewed application field of xenogeneic cells.

Point 3: To support the assumption that MSC can function across the species barrier, the authors quote a paper (ref 83) which is a general review in which 89 out of 94 reports involved the use of human MSC in various other species, with only 2 pig-to-mouse (of which one entailed co-transplantation of MSC with hematopoietic stem cells) and one pig-to-rat experiments. It is therefore questionable to draw meaningful conclusion regarding the relevance of porcine stem cells to the overall field of cell transplantation from this study and a much more comprehensive review of the potential of porcine MSC has actually been provided by the paper by Bharli et al. (Curr Stem Cell Res Ther. 2016 Jan; 11(1): 78–93.) .

Response 3: In response to the remark of reviewer 1, we now give a statement on the limitations of this review. We still believe it is relevant to estimate the feasibility of MSC engraftment across species barriers without immunosuppression and therefore kept it in the manuscript. However, we additionally included more specific references on porcine MSCs in lines 254-257.

As we totally agree with reviewer 1 that the review of Bharli et al. is comprehensive and well written, we included the reference with pleasure in lines 258-259.

Point 4: More importantly, the case is made that porcine MSC could have the advantage of an unlimited scale-up. The authors state that "porcine stem cells of various origin can be harvested in advance at all times". While this is true, this advantage is now shared by pluripotent stem cells and, more particularly, induced pluripotent stem cells whose growth potential and suitability for large-scale banking challenge those of porcine cells (while largely eliminating the risks of pig-specific infection). Unfortunately, this issue is not addressed in the manuscript.

Response 4: We thank the reviewer for this valuable input. Following the suggestion, pluripotent stem cells as well as the option of cell banking are now addressed in the manuscript and discussed in lines 436-450.

Point 5: The authors also state that in the case of acute myocardial infarction, time constraints result in that one cannot rely on, "freshly isolated cells of allogeneic donors". This is not true as cryopreserved allogeneic cells can be rapidly thawed and used.

Response 5: As stated completely right from reviewer 1, cryopreservation is indeed a common way to provide cells for acute treatments and thus is now stated as an option in lines 411-412. Additionally, we inserted a discussion on potential drawbacks of cryopreservation on MSC functionality in lines 415-422.

Point 6: It is also stated that the porcine extracellular matrix could be "ideal as scaffold for cardiac patches". However, there is no mention of the clinical trial which has tested this strategy (Traverse et al. JACC Basic Transl Sci. 2019 Oct; 4(6): 659–669.).

Response 6: The authors are grateful for this remark and readily included the reference in lines 470-473.

Finally, the authors want to express their gratitude for the critical comments and expert advice.

Reviewer 2 Report

Gallow et al. review the advances and potential uses of porcine stem cells to treat human diseases. There are some comments to address.

  1. The autologous/allogeneic MSC studies in pig table is missing numerous studies, including some combination cell studies (e.g., Williams et al. Circulation 127:213 (2013)& Karantalis et al. JACC 66:1990 (2016))
  2. The authors initially discuss both MSCs and cardiac progenitors (CPCs). but later focus almost exclusively on MSCs, including genetic engineering of MSCs. The potential of combination cell therapy was also not discussed.
  3. The authors discuss the therapeutic effects associated with upregulation of AKT in MSCs. More recently than the cited 2006 study, was a study by Kulandavelu et al. [J Am Coll Cardiol 68:2454 (2016)] where Pim 1 was overexpressed in c-kit+ CPCs.
  4. In the section on future prospects, the authors mention allogeneic stem cells but dismiss them due to the time needed to prepare a sufficient number of cells. This obstacle is really associated with autologous stem cells. Allogeneic stem cells can, and have been, grown to large numbers and are therefore, ready to inject into patients without the wait associated with autologous MSCs. The use of other, allogeneic stem cells has not been explored as extensively (e.g. Natsumeda et al. (Ref #103)). Please address if and how this issue relates to the use of (humanized) porcine cells

Author Response

Point 1: The autologous/allogeneic MSC studies in pig table is missing numerous studies, including some combination cell studies (e.g., Williams et al. Circulation 127:213 (2013)& Karantalis et al. JACC 66:1990 (2016))

Response 1: We thank the reviewer for suggesting these interesting papers that helped underlining the potential of combinatorial therapy also in xenogeneic settings as now stated in lines 268 -272.  However, since both studies investigated the effect of human CSC and MSC and only utilized the pig as model system, these studies were not included in the table summarizing studies on porcine MSC.

Point 2: The authors initially discuss both MSCs and cardiac progenitors (CPCs). but later focus almost exclusively on MSCs, including genetic engineering of MSCs. The potential of combination cell therapy was also not discussed.

Response 2: We consider MSCs as the most promising cell type for xenogeneic therapeutic applications and thus focus on them intentionally. Unfortunately, there were no multiple studies on combination cell therapy utilizing porcine cells. However, we now explicitly refer to this option and its potential in lines 268 -272 instead of just listing the only respective study with porcine cells in the table.

Point 3: The authors discuss the therapeutic effects associated with upregulation of AKT in MSCs. More recently than the cited 2006 study, was a study by Kulandavelu et al. [J Am Coll Cardiol 68:2454 (2016)] where Pim 1 was overexpressed in c-kit+ CPCs.

Response 3: We thank the reviewer for hinting at this publication. In fact, through this we noticed that the respective heading of the section doesn´t imply that we further focus on MSCs. The heading is now edited accordingly. Unfortunately, the reference could not be included, as it investigates the effect of Pim1 overexpression in human CPCs and therefore is a bit out of focus.

Point 4: In the section on future prospects, the authors mention allogeneic stem cells but dismiss them due to the time needed to prepare a sufficient number of cells. This obstacle is really associated with autologous stem cells. Allogeneic stem cells can, and have been, grown to large numbers and are therefore, ready to inject into patients without the wait associated with autologous MSCs. The use of other, allogeneic stem cells has not been explored as extensively (e.g. Natsumeda et al. (Ref #103)). Please address if and how this issue relates to the use of (humanized) porcine cells

Response 4: The availability of allogeneic stem cells due to cryopreservation is now correctly addressed and discussed in lines 411-422. As mentioned in “Response 2” we would prefer to keep the focus on MSCs. However, in lines 436- 450 we inserted a paragraph on the use of another allogeneic stem cell type, namely allogeneic iPSCs and discussed potential drawbacks that would also apply to humanized porcine cells.

Finally, we want to thank for the valuable input.

Reviewer 3 Report

Conceptual novelty

As the field of regenerative medicine becomes invested in cell therapy and since MSCs are highly acknowledged as a critical cell source for therapeutic purposes, reviews such as this one become timely appropriate. This review can be informative to the readers who seek an overview of porcine xenotransplantation products including MSCs and their therapeutic applications.

Significance to the field

This review will be informative and helpful for readers who are looking for information regarding the progress of porcine xenotransplantation. However, the future directions of these studies need to be discussed in the manuscript.

Impact upon current understanding

This review summarizes the general knowledge on porcine xenotransplantation products as a potential source for cardiac therapy. The overall scope of the review on known features and applications of porcine xenotransplantation and stem cell therapy highlights the necessity of further studying porcine xenotransplantation products and the benefit of unlocking their therapeutic potential.

Quality of writing: Overall, the review is well constructed. The authors have properly utilized figures and tables to illustrate key points throughout the manuscript. Quality of the writing is acceptable but has room for improvement. Moreover, the review lacks elaboration upon the topics listed below. This information would provide the readers with a better insight on the highlighted problem, proposed solution and the current progress in the field.

Page 1, lines 40-41: Please include detailed information regarding the time (days) it takes to prepare autologous cells in existing protocols as well as the timeline (days) for early inflammatory and later reparative phases post injury.

Page 2, lines 46-47: Describe the anatomical and physiological features that make pigs the most suitable candidates as source for xenogeneic therapies.

Section 2.1: Treatments to prevent rejection must be described in more details.

Page 2, lines 65-66:  The sentence “As the causative factors of this hyperacute rejection were considered similar to ABO-65 incompatibility in allograft rejection, similar measures were applied to prevent rejection, for example the depletion of anti-Gal antibodies by immunoaffinity columns” needs to be rewritten for clarity.

Page 3, lines 94-96: Cited paper needs to be described in more details. For example, what they mean by “prolonged graft survival”. This is a typical issue throughout the manuscript. Cited papers need to be discussed by the authors rather than solely reported. For instance, what are the downfalls of genetically engineering pigs to overcome rejection?

Page 5, lines 220-221: “several studies provided evidence that MSC successfully engrafted and functioned across the species barrier without any immunosuppression”. Please cite the original research articles as opposed to a review paper.

Section 4.1: Only one xenogenic study is listed in table 1. Almost all the studies on the effect of porcine MSCs in heart regeneration are either autologous or allogeneic. Their application in xenogenic therapy is not well assessed and this limitation needs to be acknowledged and discussed in the manuscript.

Define “ATMSCs”, “ATSCs”, “CDCs” at the first use.

There are two sections labeled “4.2.”

Are the conclusions and determinations accurate?

Conclusions are accurately based on the reported findings. However, the review could be improved through more rigorous discussion on the limitations of the proposed approaches including the time cost and financial burden of engineering pigs and viral elimination strategies as well as the topics remained to be explored for further improvement of porcine xenotransplantation studies.

Author Response

Point 1: Conceptual novelty

As the field of regenerative medicine becomes invested in cell therapy and since MSCs are highly acknowledged as a critical cell source for therapeutic purposes, reviews such as this one become timely appropriate. This review can be informative to the readers who seek an overview of porcine xenotransplantation products including MSCs and their therapeutic applications.

Response 1: We totally agree with reviewer 3 on the relevance of this field and thank for the feedback. We also want to thank for the comment, since in fact it was the basic idea of our review to develop an overview of porcine xenotransplantation in CV research.

Point 2: Significance to the field

This review will be informative and helpful for readers who are looking for information regarding the progress of porcine xenotransplantation. However, the future directions of these studies need to be discussed in the manuscript.

Response 2: We thank the reviewer for the evaluation. To enable a realistic outlook on future directions we added a small discussion with respect to the recent challenging of porcine cell therapy products by iPSCs in lines 436-450 and the drawback of the current gene editing strategy in lines 427-433. Moreover, we extended the paragraph on porcine decellularized tissues in lines 470-473 and subsequently indicated assumptions about the future directions in lines 474-475 as well as lines 433-435.

Point 3: Impact upon current understanding

This review summarizes the general knowledge on porcine xenotransplantation products as a potential source for cardiac therapy. The overall scope of the review on known features and applications of porcine xenotransplantation and stem cell therapy highlights the necessity of further studying porcine xenotransplantation products and the benefit of unlocking their therapeutic potential.

Response 3: As before, we thank the reviewer for the feedback. Indeed, we tried to outline the unabated relevance of studies on porcine xenotransplantation products despite the recent shift of focus towards other cell and matrix sources.

Point 4: Quality of writing

Overall, the review is well constructed. The authors have properly utilized figures and tables to illustrate key points throughout the manuscript. Quality of the writing is acceptable but has room for improvement. Moreover, the review lacks elaboration upon the topics listed below. This information would provide the readers with a better insight on the highlighted problem, proposed solution and the current progress in the field.

Response 4: As non-native speakers, we are sure that there is room for improvement and we will be happy to implement any suggestions from either the reviewer or the editor. We attempted to improve the writing in particular by shortening of some sentences. See for example lines 40, 220-222 or 469-470

We addressed all topics listed below to improve the value of this review for the readers.

Point 5: Page 1, lines 40-41

Please include detailed information regarding the time (days) it takes to prepare autologous cells in existing protocols as well as the timeline (days) for early inflammatory and later reparative phases post injury.

Response 5: According to the recommendation, we included more detailed information on the time frame of the inflammatory and reparative phase in lines 31-37 and the expansion time for autologous stem cells including a respective reference in lines 45-46.

Point 6: Page 2, lines 46-47

Describe the anatomical and physiological features that make pigs the most suitable candidates as source for xenogeneic therapies.

Response 6: Following the request of the reviewer, we explicitly stated the features that render the pig a suitable candidate especially regarding the cardiovascular system and included a respective reference in lines 52-57.

Point 7: Section 2.1:

Treatments to prevent rejection must be described in more details.

Response 7: As requested, mechanisms of action are now mentioned and underlined with relevant examples including general administration regimens in lines 97-106.

Point 8: Page 2, lines 65-66

The sentence “As the causative factors of this hyperacute rejection were considered similar to ABO-65 incompatibility in allograft rejection, similar measures were applied to prevent rejection, for example the depletion of anti-Gal antibodies by immunoaffinity columns” needs to be rewritten for clarity.

Response 8: We are thankful for the input and rewrote the sentence to a small paragraph in lines 78-86. To clarify the reference, the respective study is now concretely outlined and the given example is stated in more detail.

Point 9: Page 3, lines 94-96:

Cited paper needs to be described in more details. For example, what they mean by “prolonged graft survival”. This is a typical issue throughout the manuscript. Cited papers need to be discussed by the authors rather than solely reported. For instance, what are the downfalls of genetically engineering pigs to overcome rejection?

Response 9: According to the reviewers suggestion we included more details and points of discussion throughout the whole manuscript. For example, in line 118-129 details on the cited papers are now given and set in context. Furthermore, we discuss some downfalls in lines 427-435 and 451-455.

Point 10: Page 5, lines 220-221

“several studies provided evidence that MSC successfully engrafted and functioned across the species barrier without any immunosuppression”. Please cite the original research articles as opposed to a review paper.

Response 10: Following the recommendation of the reviewer, we now cited original research articles on porcine MSC transplantation in lines 256-257 in addition to the review article. We kept the review since it provides a nice systematic overview on the topic. However, we included a statement on its limitations in lines 254-257 and complemented it with another review in lines 258-259.

Point 11: Section 4.1

Only one xenogenic study is listed in table 1. Almost all the studies on the effect of porcine MSCs in heart regeneration are either autologous or allogeneic. Their application in xenogenic therapy is not well assessed and this limitation needs to be acknowledged and discussed in the manuscript.

Response 11: We are grateful for the constructive comment. Accordingly, we included a statement on the lack of xenogeneic studies for this application and discussed it in lines 274-282.

Point 12: Define “ATMSCs”, “ATSCs”, “CDCs” at the first use.

Response 12: We thank the reviewer for pointing at this. The abbreviations are now explained in the table heading as well as in the table of abbreviations.

Point 13: There are two sections labeled “4.2.”

Response 13: We want to thank the reviewer for making us aware of this mistake and corrected it accordingly.

Point 14: Are the conclusions and determinations accurate?

Conclusions are accurately based on the reported findings. However, the review could be improved through more rigorous discussion on the limitations of the proposed approaches including the time cost and financial burden of engineering pigs and viral elimination strategies as well as the topics remained to be explored for further improvement of porcine xenotransplantation studies.

Response 14: We are grateful for the critical feedback of reviewer 3 that helped us to improve the manuscript. To increase the relevance of this review, we included several points of discussion including the lack of studies on porcine xenotransplantation for cell therapy in lines 274-282, the challenge of porcine MSCs by human iPSCS in lines 436- 450 and the limitations introduced by costs and husbandry challenges in lines 451-456. Moreover, we attempted to explore the opportunities for further studies as in lines 419-422, 433-435, and 470-475.

The authors wan to explicitly thank this reviewer for many helpful and expert suggestions.

Round 2

Reviewer 1 Report

The authors have scholarly addressed the various concerns. However, it is inappropriate to state that the growth of autologous cells takes 3 weeks as this duration varies with the cell type.

Reviewer 2 Report

The authors addressed most of my concerns and improved the manuscript. In their response to Point #2, they state, "Unfortunately, there were no multiple studies on combination cell therapy utilizing porcine cells". However, there are two such studies, Karantalis et al. (Ref #109) and Natsumeda et al. (Ref #107).

Minor comment:

The term "genetical engineering" should read "genetic engineering"